# Exploring Single-Cell Data with Deep Multi-tasking Neural Networks

**Matthew Amodio, Krishnan Srinivasan, David van Dijk,**
**Hussein Mohsen, Kristina Yim, Rebecca Muhle, Kevin R. Moon**,
**Susan Kaech, Ryan Sowell, Ruth Montgomery, James Noonan**,
**Guy Wolf, & Smita Krishnaswamy**
Yale University
New Haven, CT 06520
{matthew.amodio,krishnan.srinivasan,david.vandijk,
hussein.mohsen, kristina.yim,rebecca.muhle,kevin.moon,
susan.kaech,ryan.sowell,ruth.montgomery,james.noonan,
guy.wolf,smita.krishnaswamy}@yale.edu

## Abstract

Handling the vast amounts of single-cell RNA-sequencing and CyTOF data, which are now being generated in patient cohorts, presents a computational challenge due to the noise, complexity, sparsity and batch effects present. Here, we propose a unified deep neural network-based approach to automatically process and extract structure from these massive datasets. Our unsupervised architecture, called SAUCIE (Sparse Autoencoder for Unsupervised Clustering, Imputation, and Embedding), simultaneously performs several key tasks for single-cell data analysis including 1) clustering, 2) batch correction, 3) visualization, and 4) denoising/imputation. SAUCIE is trained to recreate its own input after reducing its dimensionality in a 2-D embedding layer which can be used to visualize the data. Additionally, it uses two novel regularizations: (1) an information dimension regularization to penalize entropy as computed on normalized activation values of the layer, and thereby encourage binary-like encodings that are amenable to clustering and (2) a Maximal Mean Discrepancy penalty to correct batch effects. Thus SAUCIE has a single architecture that denoises, batch-corrects, visualizes and clusters data using a unified representation. We show results on artificial data where ground truth is known, as well as mass cytometry data from dengue patients, and single-cell RNA-sequencing data from embryonic mouse brain.

## 1 Introduction

Vast amounts of high-dimensional, high-throughput, single-cell data measuring various aspects of cells including mRNA molecules, proteins, epigenetic marks and histone modifications are being generated via new technologies. Furthermore, the number of patients included in large-scale studies of single-cell data for comparing across populations or disease conditions is rapidly increasing. Processing data of this dimensionality and scale is an inherently difficult prospect, especially considering the degree of noise, batch effects, artifacts, sparsity and heterogeneity in the data. Here, we propose a deep learning approach to process and analyze this type of data (single-cell data from a cohort of patients).

While traditional deep learning methods aim to automate predictive and generative tasks, we utilize deep learning in an exploratory fashion, to reveal the structure of multi-sample data without supervision. We base our approach on the *autoencoder*. An autoencoder is a neural network that learns to recreate its own input via a low-dimensional bottleneck layer that learns meaningful representations of the data and enables a denoised recreation of the input Bengio et al. (2013); Vincent et al. (2008); Wang et al. (2014); Hinton & Salakhutdinov (2006). If the low-dimensional bottleneck is chosen to be 2-D, then it naturally serves as a visualization of these meaningful representations as well. Since neural networks learn their own features, they can reveal structure that other 2-D visualization

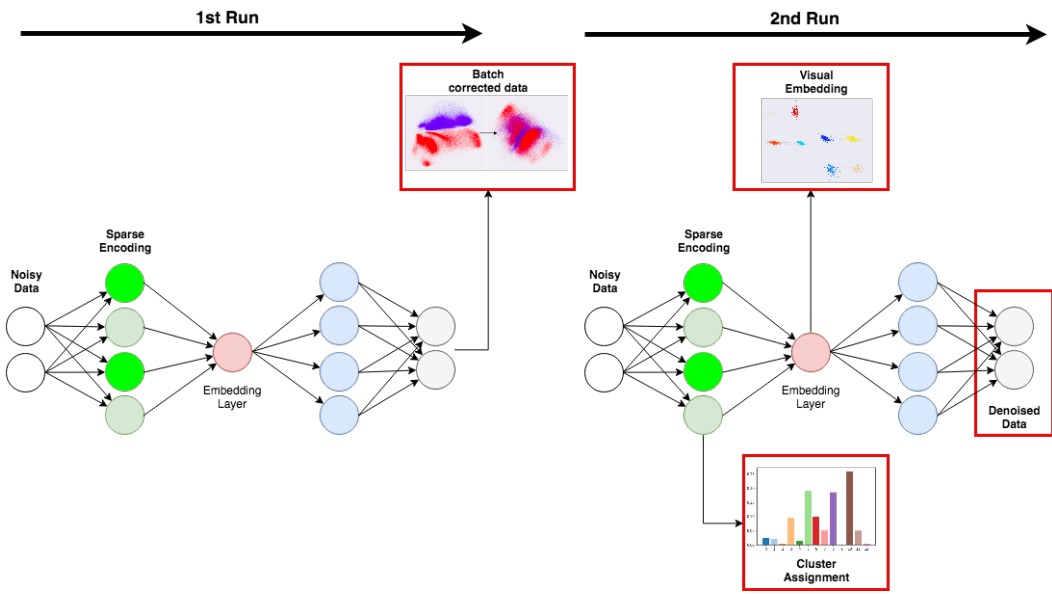

Figure 1: SAUCIE's neural network framework.

methods that require a definition of distance in the original data space cannot Maaten & Hinton (2008).

SAUCIE leverages the ability of an autoencoder to denoise, impute, and visualize, and adds carefully-designed regularizations to perform batch correction and clustering, which are essential tasks in single-cell data analysis. We introduce two novel regularizations: 1) *information dimension regularization* which allows us to recover cluster structure in a hidden layer of the autoencoder, and 2) a maximal mean discrepancy regularization that constrains the embedding layers such that different samples (of the same experimental condition) overlap with each other to perform batch correction. SAUCIE, like other modern neural networks, is also scalable to large datasets due to its massive parallelizability Simonyan & Zisserman (2014).

## 2 RESULTS

### 2.1 SAUCIE ARCHITECTURE

SAUCIE is based on the autoencoder neural network framework for unsupervised learning, which learns to reconstruct its input after passing it through a low-dimensional bottleneck layer. The bottleneck layer forces the autoencoder to learn compressed representations of the input and find high-level relationships between parts of the input space. The overall structure of an autoencoder can be divided into two parts: an encoder network that maps the input space to the low-dimensional representation, and a decoder network that maps the low-dimensional representation back to the original input space. The general architecture of SAUCIE (Figure 1) is as follows:

1. An input layer
2. A variable number of encoder layers
3. A sparse encoder layer for clustering
4. An embedding layer for visualization
5. A corresponding decoder symmetric to the encoder
6. An output layer for the reconstruction.

Different layers of SAUCIE are used for different analysis tasks. A sparse encoder layer for unsupervised clustering, an embedding layer for visualization and batch normalization, and the output layer

for reconstruction of denoised and imputed input values. Thus SAUCIE uses a unified representation and framework for many tasks, thereby increasing the coherence between tasks. For instance, the visualization and clustering in SAUCIE correspond with each other as they are in subsequent layers. We developed and implemented novel regularizations designed to restrict the representations in particular layers to achieve particular tasks. These regularizations used together form a pipeline for using neural networks to analyze biological data.

## 2.2 MULTITASK TRAINING BY SEQUENTIAL OPTIMIZATION

To perform multiple tasks, SAUCIE uses a single architecture, but is run and optimized twice sequentially. The first run imputes noisy values and corrects batch effects in the original data, while also providing two-dimensional coordinates for visualization. This preprocessed data is then run through SAUCIE again to pick out clusters. The two different runs are done by optimizing two different objective functions.

### 2.2.1 MMD REGULARIZATION FOR BATCH CORRECTION

A major challenge in the analysis of single-cell data is dealing with so-called batch effects, which result from technical variability between replicates of an experiment. Combining replicates often results in technical and experimental artifacts being the dominant source of variability in the data, even though this variability is entirely artificial. This experimental noise can come in the form of dropout, changes of scale, changes of location, or even more complicated differences in the distributions of each batch. It is infeasible to parametrically address all of the potential differences explicitly, such as assuming measurements are drawn from a Gaussian distribution. Instead of addressing specific models of noise, SAUCIE minimizes a distance metric between distributions, the Maximal Mean Discrepancy Dziugaite et al. (2015) (MMD) between batches.

### 2.2.2 INFORMATION DIMENSION REGULARIZATION FOR CLUSTERING

In order to automatically learn an appropriate granularity of clusters, we developed a novel regularization that encourages near-binary activations and minimizes the information (i.e., number of clusters) in the clustering layer. Our regularization is inspired by the von Neumann (or spectral) entropy of a linear operator Anand et al. (2011), which is computed as the Shannon entropy of their normalized eigenvalues Arpit et al. (2016); Glorot et al. (2011). This entropy serves as a proxy for the numerical rank of the operator Moon et al. (2017), and thus provides an estimation of the essential dimensionality of its range. In our case, we extend this notion to the nonlinear transformation of the neural network by treating neurons as our equivalent of eigenvalues, and computing the entropy of their total activation over a batch. We call this entropy 'information dimension' (ID) and the corresponding ID regularization aims to minimize this entropy while still encoding sufficient information to allow reconstruction of the input data points.

### 2.2.3 INTRACLUSTER DISTANCE REGULARIZATION

SAUCIE learning digital codes creates an opportunity to interpret them as clusters, but these clusters wouldn't necessarily be comprised of only similar points. To emphasize that inputs only be represented by the same digital code if they are similar to each other, SAUCIE also penalizes intracluster pairwise distances. Beyond suffering reconstruction loss, using the same code for points that are far away from each other will now incur an even greater loss.

## 3 DISCUSSION

Deep neural networks have been shown to be effective in processing massive datasets such as in the context of images and search engines. Here we apply, for the first time, a deep learning approach in an unsupervised fashion to large-scale single-cell data. As single-cell datasets become larger both in terms of cells and experimental samples (batches), scalability of analysis methods becomes key. The GPU-based parallelizable approach that training of neural networks offer makes deep learning especially suited for finding structure in single-cell data. We have presented SAUCIE, a new autoencoder framework, that performs four key tasks in single-cell data analysis: 1) data imputation, 2) clustering, 3) batch correction, 4) visualization.

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
