# OpenReview forum: "Exploring Single-Cell Data with Deep Multitasking Neural Networks"
_ICLR.cc/2018/Workshop — Reject_

### Official Review · AnonReviewer1 · 2018-02-28
**Unclear presentation.**

**Rating:** 3
**Confidence:** 3

**Review:**

The authors propose a deep auto-encoder with "information dimension" regularisation on the bottleneck layer and an additional "maximal mean discrepancy" regularisation.

It is unfortunately not clear from the paper how exactly the proposed technique works. The authors are telling what is done, but not how it is done. In particular, there are no formulas and it is not clear what exactly is being computed or optimised. It is not clear how the distance between batches is minimized and not clear how information dimension regularisation is applied.

Abstract says "we show results", but no results are show in the paper.
Section 2, called "Results", describes the proposed technique, but does not show any actual results.

The problem of cell data analysis is important, but it is not clear how the proposed technique works in practice.

---

### Official Review · AnonReviewer3 · 2018-03-04
**Needs clarity in writing and some results to properly understand the proposed framework**

**Rating:** 6
**Confidence:** 5

**Review:**

This paper uses deep autoencoders for an important biological problem. It uses standard applications of autoencoders such as dimension reduction and denoising. The novelty claim is around two new regularization techniques that are useful for the application. But without a better understanding of the data set and some preliminary result, it is difficult to understand the proposed framework. The paper should also be written more clearly to help readers understand the contributions better.

---

### Official Review · AnonReviewer2 · 2018-03-04
**SAUCIE**

**Rating:** 3
**Confidence:** 5

**Review:**

The paper presents a unified DL architecture for clustering, imputing, batch correcting and visualizing large datasets. With  no equations and experimental figures, it is difficult to understand why SAUCIE works.

> Although the authors mention in the abstract that they have results of SAUCIE on different data sets, there were none presented in the paper.

> There is no mention of how imputing/denoising happens.

> Using MMD for batch correction is a good idea but how would you discriminate for batches that have near-identical MMDs?

> Section 2.2.2 is not clearly explained. What does ‘treating neurons as our equivalent of eigenvalues’ mean? What is the dimension of the data the Shannon entropy is applied upon? Is the information this gives you (i.e. number of clusters) robust?

> Section 2.2.3 - How are you penalizing intracluster distances?

---

### Decision · Program_Chairs · 2018-03-20
**ICLR 2018 Workshop Acceptance Decision**

**Decision:**

Reject

**Comment:**

Based on the reviews, this paper has not been accepted for presentation at the ICLR workshop. However, the conversation and updates can continue to appear here on OpenReview.